# Fair principal component analysis (PCA): minorization-maximization algorithms for Fair PCA, Fair Robust PCA and Fair Sparse PCA

**Prabhu Babu**                                                     *Prabhu.Babu@care.iitd.ac.in*
*Centre for Applied Research in Electronics*
*Indian Institute of Technology, Delhi 110016, India*

**Petre Stoica**                                                              *ps@it.uu.se*
*Division of Systems and Control, Department of Information Technology*
*Uppsala University, Uppsala, Sweden 75237*

**Astha Saini**                                                     *astha.saini@care.iitd.ac.in*
*Centre for Applied Research in Electronics*
*Indian Institute of Technology, Delhi 110016, India*

**Reviewed on OpenReview:** *https://openreview.net/forum?id=6jTQrr3APY*

## Abstract

In this paper we propose a new iterative algorithm to solve the fair PCA (FPCA) problem. We start with the max-min fair PCA formulation originally proposed in Samadi et al. (2018) and derive a simple and efficient iterative algorithm which is based on the minorization-maximization (MM) approach. The proposed algorithm relies on the relaxation of a semi-orthogonality constraint which is proved to be tight at every iteration of the algorithm. The vanilla version of the proposed algorithm requires solving a semi-definite program (SDP) at every iteration, which can be further simplified to a quadratic program by formulating the dual of the surrogate maximization problem. We also propose two important reformulations of the fair PCA problem: a) fair robust PCA - which can handle outliers in the data, and b) fair sparse PCA - which can enforce sparsity on the estimated fair principal components. The proposed algorithms are computationally efficient and monotonically increase their respective design objectives at every iteration. An added feature of the proposed algorithms is that they do not require the selection of any hyperparameter (except for the fair sparse PCA case where a penalty parameter that controls the sparsity has to be chosen by the user). We numerically compare the performance of the proposed methods with two of the state-of-the-art approaches on synthetic data sets and real-life data sets.

## 1 Introduction and problem formulations

Principal component analysis (PCA) is one of the most widely used unsupervised dimensionality reduction technique Jolliffe (2002), which has found applications, among others, in image processing Clausen & Wechsler (2000); Du & Fowler (2008), signal processing Castells et al. (2007); Subasi & Gursoy (2010), finance Tsay (2005), and biology Giuliani (2017); Lever et al. (2017). Given the data, PCA learns the underlying low dimensional subspace by minimizing the reconstruction error. Let $\mathbf{Y} = [\mathbf{y}_1, \mathbf{y}_2, \cdots, \mathbf{y}_N] \in \mathbb{R}^{n \times N}$ denote the data samples, where $n$ and $N$ ($n \leq N$) represent the dimension and the number of data samples, respectively. Without loss of generality, we assume that the data samples are mean centered, i.e. $\sum_{i=1}^{N} \mathbf{y}_i = \mathbf{0}$.

Mathematically, PCA solves the following problem to find the low rank matrix $\hat{\mathbf{Y}}$:

$$\hat{\mathbf{Y}} = \arg\min_{\mathbf{Z}} \|\mathbf{Y} - \mathbf{Z}\| \text{ s.t. } \mathrm{rank}(\mathbf{Z}) = r \leq n \tag{1}$$

where $\|\cdot\|$ denotes the Frobenius norm and $r$ denotes the rank of the low-dimensional subspace, which is usually pre-specified. Although the problem in (1) is non-convex (due to the presence of the rank constraint), its solution can be computed analytically using the singular value decomposition of $\mathbf{Y}$ :

$$\mathbf{Y} = \begin{bmatrix} \underbrace{\mathbf{U}}_{r} & \underbrace{\bar{\mathbf{U}}}_{n-r} \end{bmatrix} \begin{bmatrix} \boldsymbol{\Sigma} & 0 \\ 0 & \bar{\boldsymbol{\Sigma}} \end{bmatrix} \begin{bmatrix} \mathbf{V}^T \} \, r \\ \bar{\mathbf{V}}^T \} \, n-r \end{bmatrix}$$

where the diagonal matrix $\begin{bmatrix} \boldsymbol{\Sigma} & 0 \\ 0 & \bar{\boldsymbol{\Sigma}} \end{bmatrix}$ contains the singular values in decreasing order and the unitary matrices $\begin{bmatrix} \mathbf{U} \, \bar{\mathbf{U}} \end{bmatrix}$ and $\begin{bmatrix} \mathbf{V}^T \\ \bar{\mathbf{V}}^T \end{bmatrix}$ comprise the corresponding left and right singular vectors. Then, the low rank approximation $\hat{\mathbf{Y}}$ is given by

$$\hat{\mathbf{Y}} = \mathbf{U}\boldsymbol{\Sigma}\mathbf{V}^T = \mathbf{U}\mathbf{U}^T\mathbf{Y}$$

The reconstruction error is given by

$$\|\mathbf{Y} - \hat{\mathbf{Y}}\|^2 = \|(\mathbf{I}_n - \mathbf{U}\mathbf{U}^T)\mathbf{Y}\|^2 \begin{aligned} &= \mathrm{Tr}[\mathbf{Y}^T(\mathbf{I}_n - \mathbf{U}\mathbf{U}^T)\mathbf{Y}] \\ &= \mathrm{const.} - \mathrm{Tr}(\mathbf{U}^T\mathbf{R}\mathbf{U}) \end{aligned} \tag{2}$$

where $\mathbf{R} \triangleq \mathbf{Y}\mathbf{Y}^T$ is the sample covariance matrix.
Using (2), the problem in (1) can be reformulated as:

$$\max_{\mathbf{U}} \quad \mathrm{Tr}(\mathbf{U}^T\mathbf{R}\mathbf{U}) \text{ s.t. } \mathbf{U}^T\mathbf{U} = \mathbf{I}_r \tag{3}$$

The maximizer of (3) is given by the $r$ principal eigenvectors of $\mathbf{R}$, and the cost function in (3) is the Rayleigh quotient which can also be interpreted as the projection variance. However, in real-life applications the data samples come from diverse classes, so maximizing the average variance as in (3) cannot ensure fairness among the classes. In other words, the matrix $\mathbf{U}$ determined by solving (3) may be a good fit for some data samples and a poor fit for others. To mitigate this problem, a worst case approach named Fair PCA (FPCA) was recently proposed, which solves the following problem. Let us assume that the data samples are from $K$ different classes with the matrix $\mathbf{Y}_k$ containing the samples belonging to class $k$. The corresponding sample covariance matrix is $\mathbf{R}_k \triangleq \mathbf{Y}_k\mathbf{Y}_k^T$. Then the FPCA problem can be stated as Samadi et al. (2018):

$$\max_{\mathbf{U}} \quad \min_{k \in [1,K]} \quad \mathrm{Tr}(\mathbf{U}^T\mathbf{R}_k\mathbf{U}) \text{ s.t. } \mathbf{U}^T\mathbf{U} = \mathbf{I}_r \tag{4}$$

The solution to (4) will give the best fit for the least favourable class, and in general will balance the fitting with potentially many classes having identical variances $\{\mathrm{Tr}(\mathbf{U}^T\mathbf{R}_k\mathbf{U})\}$. The FPCA problem in (4) is hard to solve due to the presence of the inner minimization operator and the constraint. In Samadi et al. (2018) and Tantipongpipat et al. (2019), the authors have proposed algorithms for solving (4) using a semidefinite relaxation (SDR) technique. However the relaxation employed by SDR may not be tight for all problem dimensions and therefore may yield suboptimal solutions (except in the case $K = 2$ where the relaxation is always tight as proved in Samadi et al. (2018)). In Kamani et al. (2022), the authors have proposed an alternative formulation of FPCA as a multi-objective optimization problem and suggested an iterative method to arrive at solutions spanning the Pareto frontier. Finally Zalcberg & Wiesel (2021) has proposed a sub-gradient based algorithm to solve the FPCA problem. However, as for any gradient approach, the algorithm of Zalcberg & Wiesel (2021) requires a careful choice of the step-size: failing to choose a good step-size may result in either convergence to a suboptimal solution or a very slow convergence.

In this paper, we propose an iterative algorithm to solve the FPCA problem, which is based on the minorization-maximization (MM) approach. Unlike some of the aforementioned algorithms, the proposed algorithm monotonically increase the cost function (i.e. $\min_k\{\mathrm{Tr}(\mathbf{U}^T\mathbf{R}_k\mathbf{U})\}$) and it is guaranteed to converge. An added feature of our algorithm is that it does not require the selection of any tuning parameter.

In the presence of outliers, the solution to the problem formulated as in (4) can be significantly affected. To tackle the outliers, we propose the following robust version of the fair PCA problem (hereafter called FRPCA):

$$\max_{\mathbf{U}} \quad \min_{k \in [1,K]} \quad \|\mathbf{U}^T \mathbf{Y}_k\|_1 \quad \text{s.t.} \quad \mathbf{U}^T \mathbf{U} = \mathbf{I}_r \tag{5}$$

where $\|\mathbf{U}^T \mathbf{Y}_k\|_1$ denotes the sum of the absolute values of the elements of $\mathbf{U}^T \mathbf{Y}_k$. The $\ell_1$ norm based fitting criterion in (5) is inspired by the L1 PCA formulation proposed in Kwak (2008) which is the robust counterpart of the vanilla PCA and has been found to be quite effective in handling outliers. The proposed MM algorithm for (4) can be relatively easily adapted to solve the FRPCA problem in (6).

Finally, we also consider an extension of the proposed MM based algorithm for (4) to cases in which the sparsity of the fair PCA solution is important. To do this, we consider the following penalized problem:

$$\max_{\mathbf{U}} \quad \min_{k \in [1,K]} \quad \text{Tr}(\mathbf{U}^T \mathbf{R}_k \mathbf{U}) - \lambda \|\mathbf{U}\|_1 \quad \text{s.t.} \quad \mathbf{U}^T \mathbf{U} = \mathbf{I}_r \tag{6}$$

where $\lambda$ is a prespecified penalty that controls the sparsity of $\mathbf{U}$. The formulation in (6) is motivated by the literature on sparse PCA Zou et al. (2006); Zou & Xue (2018); d'Aspremont et al. (2004) where the $\ell_1$ norm of $\mathbf{U}$ is usually added as a penalty to the objective in (3) to promote sparsity and hence help identify the significant contributing elements of the estimated principal components. As we will show later the proposed method for (4) can be extended to deal with the problem (6) in a straightforward manner.

The main contributions of this paper can be summarized as follows:

1) We propose a hyperparameter free algorithm based on the MM approach for solving the FPCA problem. The proposed algorithm has a monotonic convergence behavior and can be efficiently implemented.

2) We present a modification of the proposed algorithm that can deal with outliers in the data. More precisely, we formulate the fair robust PCA (FRPCA) problem and show how the MM algorithm for the FPCA approach can be modified to solve the FRPCA problem.

3) We also present an extension of the proposed algorithm (abbreviated as FSPCA) that can handle a sparsity promoting penalty in the FPCA problem.

4) Finally, we present several numerical results on both synthetic data sets and real-life data sets and compare the performance of the proposed algorithm with that of two state-of-the-art approaches.

*Notations*: We use bold uppercase (e.g., $\mathbf{Y}$), bold lower case (e.g., $\mathbf{x}$) for matrices and vectors, respectively and italic letters (e.g., $x$) for scalars. The superscript $t$ of $\mathbf{x}^t$ denotes the iteration index. For any symmetric matrices $\mathbf{A}$ and $\mathbf{B}$, $\mathbf{A} \succcurlyeq \mathbf{B}$ means that $\mathbf{A} - \mathbf{B}$ is positive semidefinite. The variables $\mathbf{I}_r$ and $\mathbf{I}_n$ denote the identity matrices of dimension $r \times r$ and $n \times n$. $\mathbf{A}^{\frac{1}{2}}$ denotes the Hermitian square-root of the positive definite matrix $\mathbf{A}$ and $\text{sgn}(x)$ denotes the sign of the scalar $x$. Finally, $\|\mathbf{U}\|_1$ denotes the sum of the absolute values of the elements of the matrix $\mathbf{U}$.

A brief outline of the paper is as follows. In Section 2 we first discuss the general MM approach for a maximization problem, and then the MM approach for max-min problems. In Section 3 we present the proposed algorithm for the FPCA problem and discuss an efficient way to implement it. In Section 4 we consider an extension of the proposed method to deal with fair robust PCA and fair sparse PCA problems. In Section 5, we present several numerical simulation results on both synthetic and real-life data and compare the proposed method with two state-of-the-art methods. Finally, we conclude the paper in Section 6.

## 2 MM Primer

In this section, we first present the main steps of the MM approach for maximization problems. Later we discuss how the MM approach can be extended to deal with max-min problems, which are of interest in this paper.

### 2.1 MM for max problems

Consider the following constrained maximization problem:

$$\max_{\mathbf{x} \in \chi} f(\mathbf{x}) \tag{7}$$

where $\mathbf{x}$ is the optimization variable, $f(\mathbf{x})$ denotes the objective function and $\chi$ is a constraint set. An MM-based algorithm solves the above problem by first constructing a surrogate function $g(\mathbf{x}|\mathbf{x}^t)$ which strictly lower bounds the objective function $f(\mathbf{x})$ at the current iterate $\mathbf{x}^t$. The function $g(\mathbf{x}|\mathbf{x}^t)$ qualifies to be a surrogate function if it satisfies the following conditions:

$$g\left(\mathbf{x}|\mathbf{x}^t\right) \leq f(\mathbf{x}) \ \forall \mathbf{x} \in \chi \tag{8}$$

$$g\left(\mathbf{x}^t|\mathbf{x}^t\right) = f\left(\mathbf{x}^t\right) \tag{9}$$

Then in the second step, the surrogate function is maximized to get the next iterate i.e.,

$$\mathbf{x}^{t+1} \in \arg\max_{\mathbf{x}\in\chi} g\left(\mathbf{x}|\mathbf{x}^t\right) \tag{10}$$

The above two steps are repeated until the algorithm converges to a stationary point of the problem in (7). The surrogate function ideally should follow the shape of the objective function and be easy to maximize.

To summarize, the major steps of the MM approach are:

1. Start with a feasible $\mathbf{x}^0$ and set $t = 0$.

2. Construct a minorizing function $g\left(\mathbf{x}|\mathbf{x}^t\right)$ of $f(\mathbf{x})$ at $\mathbf{x}^t$

3. Obtain $\mathbf{x}^{t+1} \in \arg\max_{\mathbf{x}\in\chi} g\left(\mathbf{x}|\mathbf{x}^t\right)$.

4. If $\frac{|f(\mathbf{x}^t) - f(\mathbf{x}^{t+1})|}{|f(\mathbf{x}^t)|} < \epsilon$, where $\epsilon$ is some prespecified convergence threshold, exit; otherwise set $t = t+1$ and go to Step 2.

It is straightforward to show that the MM steps monotonically increase the objective at every iteration i.e.

$$f(\mathbf{x}^{t+1}) \geq g\left(\mathbf{x}^{t+1}|\mathbf{x}^t\right) \geq g\left(\mathbf{x}^t|\mathbf{x}^t\right) = f(\mathbf{x}^t)$$

The first inequality and the third equality follow from (8) and (9) and second inequality from (10).

## 2.2   MM for max-min problems

Consider the following max-min optimization problem:

$$\max_{\mathbf{x}\in\mathcal{X}} \left\{ f(\mathbf{x}) \triangleq \min_{i=1,2,\cdots,K} f_i(\mathbf{x}) \right\} \tag{11}$$

A possible surrogate function $g(\mathbf{x}|\mathbf{x}^t)$ for the max-min problem is as follows:

$$g(\mathbf{x}|\mathbf{x}^t) = \min_{i=1,2,\cdots,K} g_i(\mathbf{x}|\mathbf{x}^t) \tag{12}$$

where each $g_i(\mathbf{x}|\mathbf{x}^t)$ is a tight lower bound on $f_i(\mathbf{x})$ at $\mathbf{x}^t$. The individual surrogates $g_i(\mathbf{x})$ satisfy the following conditions:

$$g_i(\mathbf{x}^t|\mathbf{x}^t) = f_i(\mathbf{x}^t) \tag{13}$$

$$g_i(\mathbf{x}|\mathbf{x}^t) \leq f_i(\mathbf{x}) \tag{14}$$

One can easily show that the surrogate function $g(\mathbf{x}|\mathbf{x}^t)$ defined in (12) satisfies the conditions (8) and (9):

$$\begin{aligned}
g_i(\mathbf{x}|\mathbf{x}^t) \leq f_i(\mathbf{x}) \quad &\implies \min_{i=1,2,\cdots,K} g_i(\mathbf{x}|\mathbf{x}^t) \\
&\leq \min_{i=1,2,\cdots,K} f_i(\mathbf{x}) \\
&\implies g(\mathbf{x}|\mathbf{x}^t) \leq f(\mathbf{x}).
\end{aligned} \tag{15}$$

and

$$g(\mathbf{x}^t|\mathbf{x}^t) = \min_{i=1,2,\cdots,K} g_i(\mathbf{x}^t|\mathbf{x}^t) = \min_{i=1,2,\cdots,K} f_i(\mathbf{x}^t) = f(\mathbf{x}^t) \tag{16}$$

Similar to the general MM case, here also one can show that the iterates $\{\mathbf{x}^t\}$ increase the objective function $f(\mathbf{x})$ monotonically and converge to a stationary point. We refer the reader to Naghsh et al. (2019) Zhao & Palomar (2016) for application of MM to max-min problems in communications and radar, and to Sun et al. (2016) for a detailed discussion of the MM approach including different ways of finding the surrogate function in the context of different applications.

## 3 FPCA

In this section, we will derive the MM algorithm for the problem in (3). For the sake of convenience, we restate the FPCA problem:

$$\max_{\mathbf{U}} \quad \{f(\mathbf{U}) \triangleq \min_{k\in[1,K]} \ f_k(\mathbf{U})\} \ \text{s.t.} \ \mathbf{U}^T\mathbf{U} = \mathbf{I}_r \tag{17}$$

where $f_k(\mathbf{U}) \triangleq \operatorname{Tr}(\mathbf{U}^T\mathbf{R}_k\mathbf{U})$. Before we venture into solving (17), we state and prove a lemma which will help us in the development of the proposed algorithm.

*Lemma* 1. The non-convex semi-orthogonality constraint $\mathbf{U}^T\mathbf{U} = \mathbf{I}_r$ in (17) can be relaxed to $\mathbf{U}^T\mathbf{U} \preccurlyeq \mathbf{I}_r$ and the global maximizer of the relaxed problem will satisfy the constraint in (17).

*Proof.* Let the eigenvalue decomposition of $\mathbf{U}\mathbf{U}^T$ be as follows:

$$\mathbf{U}\mathbf{U}^T = \mathbf{V}\mathbf{\Lambda}\mathbf{V}^T \tag{18}$$

where $\mathbf{V}$ contains the principal eigenvectors of $\mathbf{U}\mathbf{U}^T$ and $\mathbf{V}^T\mathbf{V} = \mathbf{I}_r$. Using (18) we have :

$$\operatorname{Tr}(\mathbf{U}^T\mathbf{R}_k\mathbf{U}) = \operatorname{Tr}((\mathbf{V}^T\mathbf{R}_k\mathbf{V})\mathbf{\Lambda}) \quad = \sum_{i=1}^{r}(\mathbf{V}^T\mathbf{R}_k\mathbf{V})_{ii}\mathbf{\Lambda}_{ii}$$
$$\leq \sum_{i=1}^{r}(\mathbf{V}^T\mathbf{R}_k\mathbf{V})_{ii}$$

Therefore all functions in (17) are larger for $\mathbf{\Lambda} = \mathbf{I}_r$ than for $\mathbf{\Lambda} < \mathbf{I}_r$, and this implies that the global maximizer of (17) under the constraint $\mathbf{U}^T\mathbf{U} \preccurlyeq \mathbf{I}_r$ will satisfy the constraint in (17). Therefore the relaxation $\mathbf{U}^T\mathbf{U} \preccurlyeq \mathbf{I}_r$ does not effect the solution of (17). ∎

Using Lemma 1 we replace the problem in (17) with the following relaxed problem:

$$\max_{\mathbf{U}} \quad \min_{k\in[1,K]} \ \operatorname{Tr}(\mathbf{U}^T\mathbf{R}_k\mathbf{U}) \ \text{s.t.} \ \mathbf{U}^T\mathbf{U} \preccurlyeq \mathbf{I}_r \tag{19}$$

The constraint in (19) is convex as it can be reformulated as a linear matrix inequality (Boyd & Vandenberghe (2004)):

$$\begin{bmatrix} \mathbf{I}_r & \mathbf{U}^T \\ \mathbf{U} & \mathbf{I}_n \end{bmatrix} \succcurlyeq 0$$

However, the maximization problem in (19) is non-convex as the objective (for any $k$) is a convex quadratic function in $\mathbf{U}$ and the presence of min operator further adds to complications. We resort to the MM approach to solve (19). Following the MM steps listed in Section 2.2, each convex quadratic function in (19) can be lower bounded via its tangent hyperplane at $\mathbf{U}^t$. Doing so gives us an MM surrogate for the objective in (19). For given $\mathbf{U}^t$ and for any $k$, we have

$$f_k(\mathbf{U}) \quad = \operatorname{Tr}(\mathbf{U}^T\mathbf{R}_k\mathbf{U}) \geq \operatorname{Tr}((\mathbf{U}^t)^T\mathbf{R}_k\mathbf{U}^t) + 2\operatorname{Tr}((\mathbf{U}^t)^T\mathbf{R}_k(\mathbf{U} - \mathbf{U}^t))$$
$$= 2\operatorname{Tr}((\mathbf{U}^t)^T\mathbf{R}_k\mathbf{U}) - \operatorname{Tr}((\mathbf{U}^t)^T\mathbf{R}_k\mathbf{U}^t) \triangleq g_k(\mathbf{U})$$

Then the surrogate problem is given by:

$$\max_{\mathbf{U}} \quad \min_{k} \ g_k(\mathbf{U}) \ \text{s.t.} \ \begin{bmatrix} \mathbf{I}_r & \mathbf{U}^T \\ \mathbf{U} & \mathbf{I}_n \end{bmatrix} \succcurlyeq 0 \tag{20}$$

Using the expression for $g_k(\mathbf{U})$ in (20) we get

$$\max_{\mathbf{U}} \quad \min_{k} \quad 2\operatorname{Tr}(\mathbf{A}_k^T\mathbf{U}) + c_k \quad \text{s.t.} \quad \begin{bmatrix} \mathbf{I}_r & \mathbf{U}^T \\ \mathbf{U} & \mathbf{I}_n \end{bmatrix} \succcurlyeq 0 \tag{21}$$

where

$$\mathbf{A}_k^T \triangleq (\mathbf{U}^t)^T\mathbf{R}_k, \; c_k = -\operatorname{Tr}((\mathbf{U}^t)^T\mathbf{R}_k\mathbf{U}^t) \tag{22}$$

Problem (21) is convex and can be reformulated as an SDP :

$$\begin{aligned} &\max_{\alpha,\mathbf{U}} \; \alpha \\ &\text{s.t.} \; 2\operatorname{Tr}(\mathbf{A}_k^T\mathbf{U}) + c_k \geq \alpha \\ &\quad \begin{bmatrix} \mathbf{I}_r & \mathbf{U}^T \\ \mathbf{U} & \mathbf{I}_n \end{bmatrix} \succcurlyeq 0 \end{aligned} \tag{23}$$

which can be solved using off-the-shelf solvers like CVX Grant & Boyd (2014).

Consider the inner minimization problem in (20), which can be written as follows:

$$\min_{k} \; g_k(\mathbf{U}) = \min_{\boldsymbol{\mu}} \; \sum_{k=1}^{K} \mu_k g_k(\mathbf{U}) \tag{24}$$

$$\text{s.t.} \; \mu_k \geq 0, \boldsymbol{\mu}^T\mathbf{1} = \mathbf{1}.$$

Solving the problem on R.H.S. of (24), the solution elements of the vector $\boldsymbol{\mu}$ are such that $\mu_k$ corresponding to the minimum value of $g_k(\mathbf{U})$ is 1 and rest are 0, giving the minimum value of $g_k(\mathbf{U})$. So, the inner minimization problem in (21) can be reformulated using the auxiliary variables $\boldsymbol{\mu} = [\mu_1, \cdots, \mu_K]^T$, as shown below:

$$\begin{aligned} &\max_{\mathbf{U}} \; \min_{\boldsymbol{\mu}} \; \sum_{k=1}^{K} \mu_k \mathrm{g}_k(\mathbf{U}) \\ &\text{s.t.} \; \mu_k \geq 0, \; \sum_{k=1}^{K} \mu_k = 1 \\ &\quad \begin{bmatrix} \mathbf{I}_r & \mathbf{U}^T \\ \mathbf{U} & \mathbf{I}_n \end{bmatrix} \succcurlyeq 0. \end{aligned} \tag{25}$$

*Lemma* 2. The maximizer $\mathbf{U}$ of (21) (or equivalently (23)) satisfies the constraint $\mathbf{U}^T\mathbf{U} = \mathbf{I}_r$ at each iteration of the algorithm (not only at convergence).

*Proof.* This interesting property will be proved in what follows. Note that (25) is a notationally simpler reformulation of (23). The objective in (25) is a linear function of $\mathbf{U}$ for given $\boldsymbol{\mu}$, and it is linear in $\boldsymbol{\mu}$ for fixed $\mathbf{U}$. Furthermore the constraint sets for both $\mathbf{U}$ and $\boldsymbol{\mu}$ are compact and convex. Consequently, using the minimax theorem Sion (1958), the max and min operators in (25) can be interchanged and we get the following equivalent problem:

$$\begin{aligned} &\min_{\boldsymbol{\mu}} \; \max_{\mathbf{U}} \; 2\operatorname{Tr}(\mathbf{A}^T\mathbf{U}) + \sum_{k=1}^{K} \mu_k c_k \\ &\text{s.t.} \; \mu_k \geq 0 \; , \sum_{k=1}^{K} \mu_k = 1 \; , \begin{bmatrix} \mathbf{I}_r & \mathbf{U}^T \\ \mathbf{U} & \mathbf{I}_n \end{bmatrix} \succcurlyeq 0 \end{aligned} \tag{26}$$

where

$$\mathbf{A}(\boldsymbol{\mu}) \triangleq \sum_{k=1}^{K} \mu_k \mathbf{A}_k^T \tag{27}$$

where $\mathbf{A}(\boldsymbol{\mu})$ indicates the dependency of $\mathbf{A}$ on $\boldsymbol{\mu}$. The inner maximization problem in (26) can be solved in closed form. Consider the first term in the objective of (26). By Von-Neumann inequality Marshall et al. (1979), we have

$$\operatorname{Tr}((\mathbf{A}(\boldsymbol{\mu}))^T\mathbf{U}) \leq \sum_{i=1}^{r} \sigma_i(\mathbf{A}(\boldsymbol{\mu}))\sigma_i(\mathbf{U})$$

where $\sigma_i(\mathbf{A}(\boldsymbol{\mu}))$ and $\sigma_i(\mathbf{U})$ denote the non-zero singular values of $\mathbf{A}(\boldsymbol{\mu})$ and $\mathbf{U}$, respectively. Because $\sigma_i(\mathbf{U}) \leq 1$, it follows that

$$\mathrm{Tr}((\mathbf{A}(\boldsymbol{\mu}))^T \mathbf{U}) \leq \sum_{i=1}^{r} \sigma_i(\mathbf{A}(\boldsymbol{\mu}))$$

with the equality attained for

$$\mathbf{U}^* = \mathbf{A}(\boldsymbol{\mu})((\mathbf{A}(\boldsymbol{\mu}))^T \mathbf{A}(\boldsymbol{\mu}))^{-\frac{1}{2}} \tag{28}$$

Indeed,

$$\mathrm{Tr}((\mathbf{A}(\boldsymbol{\mu}))^T \mathbf{U}^*) = \mathrm{Tr}[((\mathbf{A}(\boldsymbol{\mu}))^T \mathbf{A}(\boldsymbol{\mu}))((\mathbf{A}(\boldsymbol{\mu}))^T \mathbf{A}(\boldsymbol{\mu}))^{-\frac{1}{2}}] = \mathrm{Tr}[((\mathbf{A}(\boldsymbol{\mu}))^T \mathbf{A}(\boldsymbol{\mu}))^{\frac{1}{2}}] = \sum_{i=1}^{r} \sigma_i(\mathbf{A}(\boldsymbol{\mu}))$$

Note that $\mathbf{U}^*$ satisfies the constraint in (17) i.e. $(\mathbf{U}^*)^T \mathbf{U}^* = \mathbf{I}_r$. Therefore, as claimed, the maximizer of (23) satisfies the constraint $\mathbf{U}^T \mathbf{U} = \mathbf{I}_r$ at each iteration. ∎

Now inserting (28) in (26) yields the following problem that remains to be solved:

$$\begin{aligned} \min_{\boldsymbol{\mu}} \quad & 2 \sum_{i=1}^{K} \sigma_i(\mathbf{A}(\boldsymbol{\mu})) + \sum_{k=1}^{K} \mu_k c_k \\ \text{s.t.} \quad & \mu_k \geq 0 \;, \sum_{k=1}^{K} \mu_k = 1 \end{aligned} \tag{29}$$

where we have stressed by notation that $\mathbf{A}$ is a function of $\boldsymbol{\mu}$. The first term in (29) is equal to 2 times the matrix nuclear norm of $\mathbf{A}(\boldsymbol{\mu})$, denoted $\|\mathbf{A}(\boldsymbol{\mu})\|_*$, and it is a convex function of $\mathbf{A}$ and hence $\{\mu_k\}$. Thus, (29) is a convex problem like (23) and can be reformulated as an SDP (see Appendix A) Recht et al. (2010). However compared to (23) the number of variables and constraints in (29) is smaller, which can be an advantage. In fact for $K = 2$, (29) can be solved as a 1-dimensional problem via a bisection method.

Once the minimizer $\boldsymbol{\mu}^*$ is obtained by solving (29), the corresponding $\mathbf{U}$ (which is also the maximizer of (23)) can be obtained as:

$$\mathbf{U}^{(t+1)} = \mathbf{A}(\boldsymbol{\mu}^*)(\mathbf{A}^T(\boldsymbol{\mu}^*)\mathbf{A}(\boldsymbol{\mu}^*))^{-\frac{1}{2}} \tag{30}$$

and it will serve as the next iterate. The MM-procedure will continue in the same manner till the iterates converge. The steps of the proposed algorithm are summarized in a pseudocode form in Algorithm 1.

---

**Algorithm 1** FPCA algorithm

---

**Input** Initial estimate $\mathbf{U}^0$, $\{\mathbf{R}_k\}_{k=1}^{K}$, and convergence threshold $\epsilon = 10^{-5}$.
Set $t = 0$.
**repeat**
    Compute $\{\mathbf{A}_k, \; c_k\}$ in (22).
    Compute $\boldsymbol{\mu}^*$ by solving (29).
    Obtain $\mathbf{U}^{t+1}$ from (30).
    Set $t = t + 1$.
**until** $\dfrac{f(\mathbf{U}^{t+1}) - f(\mathbf{U}^t)}{f(\mathbf{U}^t)} \leq \epsilon$.
$\mathbf{U}_{\mathrm{FPCA}} = \mathbf{U}^t$ at convergence.
**Output** $\mathbf{U}_{\mathrm{FPCA}}$.

---

The main burden of the proposed algorithm lies in the computation of $\{\mathbf{A}_k\}_{k=1}^{K}$, solving the SDP in (29) and computing $\mathbf{U}^{t+1}$ in (30). Computation of $\{\mathbf{A}_k\}_{k=1}^{K}$ involves evaluation of matrix-matrix products which can be done in $\mathcal{O}(Krn^2)$ flops. Solving the SDP in (29) requires roughly $\mathcal{O}((n+r)^{4.5})$ flops Shen et al. (2008); Zheng et al. (2012), and evaluating (30) requires $\mathcal{O}(nr^2) + \mathcal{O}(r^3)$ flops, thus the total number of computations per iteration is on the order of $\mathcal{O}((n+r)^{4.5})$ flops. The complexity of $\mathcal{O}((n+r)^{4.5})$ seems

to be on the higher side especially when compared to that of a recent state-of-the-art algorithm proposed in Zalcberg & Wiesel (2021), which has a complexity of $\mathcal{O}(n^3)$. Algorithm 1 can be run on a workstation for fairly large dimensions such as $n = 1000$, $r = 100$, $K = 100$. Regarding the space complexity, at every iteration, the matrices $\{\mathbf{A}_k\}_{k=1}^K$ need to be stored which requires $\mathcal{O}(Krn)$ memory words, and the space complexity of the SDP is around $\mathcal{O}((n+r)^2)$, thus the total space complexity is on the order of $\mathcal{O}((n+r)^2)$.

In the following, we propose an alternative approach to solve (29) that helps reduce the computational burden. To do so, we start with the problem in (29):

$$
\begin{aligned}
\min_{\boldsymbol{\mu}} \quad & \{\tilde{f}(\boldsymbol{\mu}) \triangleq 2\left\|\mathbf{A}(\boldsymbol{\mu})\right\|_* + \sum_{k=1}^K \mu_k c_k\} \\
\text{s.t.} \quad & \mu_k \geq 0 \ , \sum_{k=1}^K \mu_k = 1
\end{aligned}
\tag{31}
$$

Let us introduce an auxillary variable $\boldsymbol{\Phi}$ ($\boldsymbol{\Phi} \succ \mathbf{0}$) and rewrite (31) in the form:

$$
\begin{aligned}
\min_{\boldsymbol{\mu}, \boldsymbol{\Phi} \succ \mathbf{0}} \quad & \text{Tr}(\boldsymbol{\Phi}^{-1}) + \text{Tr}((\mathbf{A}(\boldsymbol{\mu}))^T \mathbf{A}(\boldsymbol{\mu})\boldsymbol{\Phi}) + \sum_{k=1}^K \mu_k c_k \\
\text{s.t.} \quad & \mu_k \geq 0 \ , \sum_{k=1}^K \mu_k = 1
\end{aligned}
\tag{32}
$$

The problems (31) and (32) are equivalent, which can be seen as follows. If we minimize (32) over $\boldsymbol{\Phi}$ for fixed $\boldsymbol{\mu}$, we get the minimizer $\boldsymbol{\Phi}^* = ((\mathbf{A}(\boldsymbol{\mu}))^T \mathbf{A}(\boldsymbol{\mu}))^{-\frac{1}{2}}$ and substituting it back in (32) we get the problem in (31). Thus, instead of solving (31) one can solve (32) to obtain the minimizer of (31). The problem in (32) can be solved using an alternating minimization approach : for given $\boldsymbol{\mu}$, say $\boldsymbol{\mu}^t$, we get the minimizer $\boldsymbol{\Phi}^t$ as explained above. For fixed $\boldsymbol{\Phi} = \boldsymbol{\Phi}^t$, the minimizer $\boldsymbol{\mu}$ can be obtained by solving the following convex problem:

$$
\begin{aligned}
\min_{\boldsymbol{\mu}} \quad & \text{Tr}((\mathbf{A}(\boldsymbol{\mu}))^T \mathbf{A}(\boldsymbol{\mu})(\boldsymbol{\Phi}^t)) + \sum_{k=1}^K \mu_k c_k \\
\text{s.t.} \quad & \mu_k \geq 0 \ , \sum_{k=1}^K \mu_k = 1
\end{aligned}
\tag{33}
$$

which can be reformulated as a quadratic program (QP):

$$
\begin{aligned}
\min_{\boldsymbol{\mu}} \quad & \boldsymbol{\mu}^T \mathbf{Q} \boldsymbol{\mu} + \sum_{k=1}^K \mu_k c_k \\
\text{s.t.} \quad & \mu_k \geq 0 \ , \sum_{k=1}^K \mu_k = 1
\end{aligned}
\tag{34}
$$

where $[\mathbf{Q}]_{i,j} \triangleq \text{Tr}(\mathbf{A}_i^T \boldsymbol{\Phi}^t \mathbf{A}_j)$. The QP in (34) can be solved using standard solvers whose computational complexity including the update of $\boldsymbol{\Phi}^t$ is on the order of $\mathcal{O}(K^3) + \mathcal{O}(n^3)$ flops. This is much less than $\mathcal{O}((n+r)^{4.5})$ but note that, unlike the problem in (29), the problem in (33) has to be solved multiple times; fortunately usually only a few iterations are needed (typically $< 10$). There is hardly any theoretical result on this aspect, nonetheless from our numerical simulation experience we have observed that solving (33) usually indeed requires around 10 iterations (even for problem dimension $n = 100$) to achieve convergence (as can also be seen from the figures in Section 5). The iterative steps of the above alternating minimization algorithm for computing the solution $\boldsymbol{\mu}^*$ to (29) are summarized in Algorithm 2.

## 4 FSPCA and FRPCA

### 4.1 Fair Sparse PCA

In this section we present an extension of the proposed method which can handle a penalty that promotes the sparsity of the fair principal components. For convenience we repeat here the statement of the sparse FPCA problem (see (6)):

$$
\begin{aligned}
\max_{\mathbf{U}} \quad & \{h(\mathbf{U}) \triangleq \min_{k \in [1,K]} \{f_k(\mathbf{U}) - \lambda\|\mathbf{U}\|_1\}\} \\
\text{s.t.} \quad & \mathbf{U}^T \mathbf{U} = \mathbf{I}_r
\end{aligned}
\tag{35}
$$

---

**Algorithm 2** Alternating minimization approach for solving (29)

---

**Input** Initial estimate $\boldsymbol{\mu}^0$, $\{c_k, \mathbf{A}_k^T\}$ and convergence threshold $\epsilon = 10^{-5}$.
Set $t = 0$.
 **repeat**
   Compute $\boldsymbol{\Phi}^t = (\mathbf{A}^T(\boldsymbol{\mu}^t)\mathbf{A}(\boldsymbol{\mu}^t))^{-\frac{1}{2}}$.
   Obtain $\boldsymbol{\mu}^{t+1}$ by solving (34).
   Set $t = t + 1$.
 **until** $\dfrac{\tilde{f}(\boldsymbol{\mu}^{t+1}) - \tilde{f}(\boldsymbol{\mu}^t)}{\tilde{f}(\boldsymbol{\mu}^t)} \leq \epsilon$.
$\boldsymbol{\mu}^* = \boldsymbol{\mu}^t$ at convergence.
**Output $\boldsymbol{\mu}^*$.**

---

where $\lambda$ is a penalty factor that controls the sparsity of $\mathbf{U}$. Selecting $\lambda$ is a basic problem in sparse optimization literature and there is a host of methods that have been suggested for making this selection, with cross validation being often the method of choice (see, e.g. Zou et al. (2006) for selecting $\lambda$ in sparse PCA applications). Similarly to the FPCA case, we relax the semi-orthogonality constraint:

$$\max_{\mathbf{U}} \quad \min_{k \in [1,K]} \quad f_k(\mathbf{U}) - \lambda\|\mathbf{U}\|_1$$
$$\text{s.t.} \quad \mathbf{U}^T\mathbf{U} \preccurlyeq \mathbf{I}_r \implies \begin{bmatrix} \mathbf{I}_r & \mathbf{U}^T \\ \mathbf{U} & \mathbf{I}_n \end{bmatrix} \succcurlyeq 0 \tag{36}$$

For fixed $\mathbf{U} = \mathbf{U}^t$, we can minorize the quadratic functions $\{f_k(\mathbf{U})\}$ using their tangent hyperplanes, and obtain the following surrogate problem

$$\max_{\mathbf{U}} \min_{k} \quad \{g_k(\mathbf{U}) - \lambda\|\mathbf{U}\|_1\}$$
$$\text{s.t.} \quad \begin{bmatrix} \mathbf{I}_r & \mathbf{U}^T \\ \mathbf{U} & \mathbf{I}_n \end{bmatrix} \succcurlyeq 0 \tag{37}$$

where as before $g_k(\mathbf{U}) = 2\operatorname{Tr}(\mathbf{A}_k^T\mathbf{U}) + c_k$. The above problem is convex (more exactly an SDP) and can be solved via CVX. Similar to the FPCA case, the maximizer of (37) can be shown to satisfy the constraint $\mathbf{U}^T\mathbf{U} = \mathbf{I}_r$. To do so, we reformulate the problem in (37) using auxiliary variables $\mathbf{B}$ and $\boldsymbol{\mu}$ as follows:

$$\max_{\mathbf{U}} \quad \min_{\boldsymbol{\mu},\mathbf{B}} \quad \sum_{k=1}^{K} \mu_k g_k(\mathbf{U}) + \lambda \operatorname{Tr}(\mathbf{B}^T\mathbf{U})$$
$$\text{s.t. } \mu_k \geq 0 \ , \sum_{k=1}^{K} \mu_k = 1 \ , \begin{bmatrix} \mathbf{I}_r & \mathbf{U}^T \\ \mathbf{U} & \mathbf{I}_n \end{bmatrix} \succcurlyeq 0, |[\mathbf{B}]_{i,j}| \leq 1 \ \forall i,j \tag{38}$$

Minimizing (38) over $\mathbf{B}$ and $\boldsymbol{\mu}$ we get the objective in the problem (37). Thus (37) and (38) are equivalent problems. Using the minimax theorem (the objective and the constraints of (38) satisfy the required conditions), the max and min operators can be swapped:

$$\min_{\boldsymbol{\mu},\mathbf{B}} \quad \max_{\mathbf{U}} \quad \sum_{k=1}^{K} \mu_k g_k(\mathbf{U}) + \lambda \operatorname{Tr}(\mathbf{B}^T\mathbf{U})$$
$$\text{s.t. } \mu_k \geq 0 \ , \sum_{k=1}^{K} \mu_k = 1 \ , \begin{bmatrix} \mathbf{I}_r & \mathbf{U}^T \\ \mathbf{U} & \mathbf{I}_n \end{bmatrix} \succcurlyeq 0, |[\mathbf{B}]_{i,j}| \leq 1 \ \forall i,j \tag{39}$$

Using the expression for $g_k(\mathbf{U}) = 2\operatorname{Tr}(\mathbf{A}_k^T\mathbf{U}) + c_k$, (39) can be rewritten as:

$$\min_{\boldsymbol{\mu},\mathbf{B}} \quad \max_{\mathbf{U}} \quad 2\operatorname{Tr}((\mathbf{A} + \frac{\lambda}{2}\mathbf{B})^T\mathbf{U}) + \sum_{k=1}^{K} \mu_k c_k$$
$$\text{s.t. } \mu_k \geq 0 \ , \sum_{k=1}^{K} \mu_k = 1 \ , \begin{bmatrix} \mathbf{I}_r & \mathbf{U}^T \\ \mathbf{U} & \mathbf{I}_n \end{bmatrix} \succcurlyeq 0, \ |[\mathbf{B}]_{i,j}| \leq 1 \ \forall i,j \tag{40}$$

Similar to (28), the maximizer $\mathbf{U}$ of (40) can be obtained in closed form:

$$\mathbf{U}^* = (\mathbf{A} + \frac{\lambda}{2}\mathbf{B})\big((\mathbf{A} + \frac{\lambda}{2}\mathbf{B})^T(\mathbf{A} + \frac{\lambda}{2}\mathbf{B})\big)^{-\frac{1}{2}}$$

which satisfies the constraint $\mathbf{U}^T\mathbf{U} = \mathbf{I}_r$. Thus, also in the FSPCA case, the MM iterates satisfy the semi-orthogonality constraint. The pseudocode of FSPCA is summarized in Algorithm 3.

---

**Algorithm 3** FSPCA algorithm

---

**Input** Initial estimate $\mathbf{U}^0$, $\{\mathbf{R}_k\}$, $\lambda$, and convergence threshold $\epsilon = 10^{-5}$.
Set $t = 0$.
 **repeat**
 Compute $\{\mathbf{A}_k, \ c_k\}$ in (22).
 Obtain $\mathbf{U}^{t+1}$ by solving (37).
 Set $t = t + 1$.
 **until** $\dfrac{h(\mathbf{U}^{t+1}) - h(\mathbf{U}^t)}{h(\mathbf{U}^t)} \leq \epsilon$.
$\mathbf{U}_{\text{SFPCA}} = \mathbf{U}^t$ at convergence.
**Output** $\mathbf{U}_{\text{SFPCA}}$.

---

## 4.2 Fair Robust PCA

We next discuss how the proposed approach can be modified to solve the fair robust PCA formulation stated in (5):

$$\max_{\mathbf{U}} \quad \{h_r(\mathbf{U}) \triangleq \min_{k \in [1,K]} \quad \|\mathbf{U}^T\mathbf{Y}_k\|_1\} \ \text{ s.t. } \ \mathbf{U}^T\mathbf{U} = \mathbf{I}_r$$

Similar to what we have done before, we relax the semi-orthogonality constraint:

$$\begin{aligned} \max_{\mathbf{U}} \ \min_{k \in [1,K]} \quad & \|\mathbf{U}^T\mathbf{Y}_k\|_1 \\ \text{s.t.} \quad & \begin{bmatrix} \mathbf{I}_r & \mathbf{U}^T \\ \mathbf{U} & \mathbf{I}_n \end{bmatrix} \succcurlyeq 0 \end{aligned} \tag{41}$$

Compared to the previous cases, we have a different data fitting function here. Nonetheless, $\|\mathbf{U}^T\mathbf{Y}_k\|_1$ can be minorized as follows. For a given $\mathbf{U}^t$,

$$\begin{aligned} \|\mathbf{U}^T\mathbf{Y}_k\|_1 \ &= \sum_{i,j} |(\mathbf{U}^T\mathbf{Y}_k)_{ij}| \geq \sum_{i,j} (\mathbf{U}^T\mathbf{Y}_k)_{ij} \frac{((\mathbf{U}^t)^T\mathbf{Y}_k)_{ij}}{|((\mathbf{U}^t)^T\mathbf{Y}_k)_{ij}|} \\ &= \sum_{i,j} (\mathbf{U}^T\mathbf{Y}_k)_{ij}\,\text{sgn}(((\mathbf{U}^t)^T\mathbf{Y}_k)_{ij}) \\ &= \text{Tr}(\mathbf{U}^T\mathbf{Y}_k\mathbf{W}_k) \end{aligned} \tag{42}$$

where

$$\mathbf{W}_k \triangleq \text{sgn}((\mathbf{U}^t)^T\mathbf{Y}_k) \tag{43}$$

Thus the surrogate problem corresponding to (41) is given by:

$$\begin{aligned} \max_{\mathbf{U}} \ \min_{k} \quad & \text{Tr}(\mathbf{U}^T\mathbf{Y}_k\mathbf{W}_k) \\ \text{s.t.} \quad & \begin{bmatrix} \mathbf{I}_r & \mathbf{U}^T \\ \mathbf{U} & \mathbf{I}_n \end{bmatrix} \succcurlyeq 0 \end{aligned} \tag{44}$$

which is once again an SDP (after rewriting in epigraph form) that can be solved via CVX. Using a similar argument to that employed in the cases of FPCA and FSPCA, here too one can show that the solution of (44) satisfies the semi-orthogonality constraint. The pseudocode of the FRPCA is summarized in Algorithm 4.

We conclude this section with a number of remarks on the proposed algorithms.

---

**Algorithm 4** FRPCA algorithm

---

**Input** Initial estimate $\mathbf{U}^0$, $\{\mathbf{R}_k\}$, $\lambda$, and convergence threshold $\epsilon = 10^{-5}$.

Set $t = 0$.

**repeat**

  Compute $\{\mathbf{W}_k\}$ in (43).

  Compute $\mathbf{U}^{t+1}$ by solving (44).

  Set $t = t + 1$.

  **until** $\dfrac{h_r(\mathbf{U}^{t+1}) - h_r(\mathbf{U}^t)}{h_r(\mathbf{U}^t)} \leq \epsilon$.

$\mathbf{U}_{\text{FRPCA}} = \mathbf{U}^t$ at convergence.

**Output** $\mathbf{U}_{\text{FRPCA}}$.

---

- The value of $\lambda$ in the case of FSPCA and the rank $r$ for all the algorithms are assumed to be prespecified. When $\lambda$ and $r$ are not specified they can be chosen using model selection approaches or cross validation.

- In the case of FSPCA and FRPCA, although we did not prove that we can relax the semi-orthogonality constraint in the original problem formulations (like we did for FPCA), we showed that the corresponding surrogate problems tightly satisfy the semi-orthogonality constraint, which means that their solutions are feasible for the original problems.

- If desired a sparsity inducing penalty can also be added to FRPCA problem and an MM algorithm can be developed.

- In the case of FSPCA and FRPCA, QPs similar to (34) can be derived and solved instead of solving the SDPs in (37) and (44).

- Regarding the convergence of the proposed methods, by the properties of MM the objective function is monotonically increasing at each iteration. Then, because the cost functions of FPCA, FSPCA and FRPCA are bounded above, the convergence of the proposed algorithms follows from the general results proved in Sun et al. (2016).

- Regarding the initialization ($\mathbf{U}^0$) for the proposed methods, in the case of FPCA and FSPCA we observed in the numerical experiments that the objective functions were often unimodal. So, we have used random semi-orthogonal matrices to initialize the algorithms. On the other hand, the objective function of FRPCA appeared to be multi-modal (see the discussion in Section 5.4), thus we have initialized the FRPCA algorithm with the FPCA solution.

- Finally we note that RPCA and SPCA are important problems in their own right and a multitude of algorithms have been suggested for solving them (see, e.g. Kwak (2008); Zou et al. (2006); Zou & Xue (2018); d'Aspremont et al. (2004)). Our algorithms FRPCA and FPSCA with $K = 1$ can also be used to solve the RPCA and SPCA problems.

## 5 Numerical simulation results

In this section, we will compare the performance of the proposed FPCA method with the SDR method of Samadi et al. (2018) and the sub-gradient method from Zalcberg & Wiesel (2021). The SDR approach uses the parametrization $\mathbf{P} = \mathbf{U}\mathbf{U}^T$ and relaxes the rank$(\mathbf{P}) = r$ constraint to $\text{Tr}(\mathbf{P}) = r$ and the semi-orthogonality constraint to $\mathbf{0} \preceq \mathbf{P} \preceq \mathbf{I}_n$. For $K = 2$, the aforementioned relaxation is tight but for $K \geq 3$ this is not necessarily true and the method often yields solutions with rank$(\mathbf{P}) > r$. Being a relaxation the objective value attained by SDR is an upper bound which is not always achievable. We first perform simulations based on synthetic data sets and later include simulation results for real-life data sets. We will also present numerical simulation results showing the performance of the proposed FSPCA and FRPCA methods.

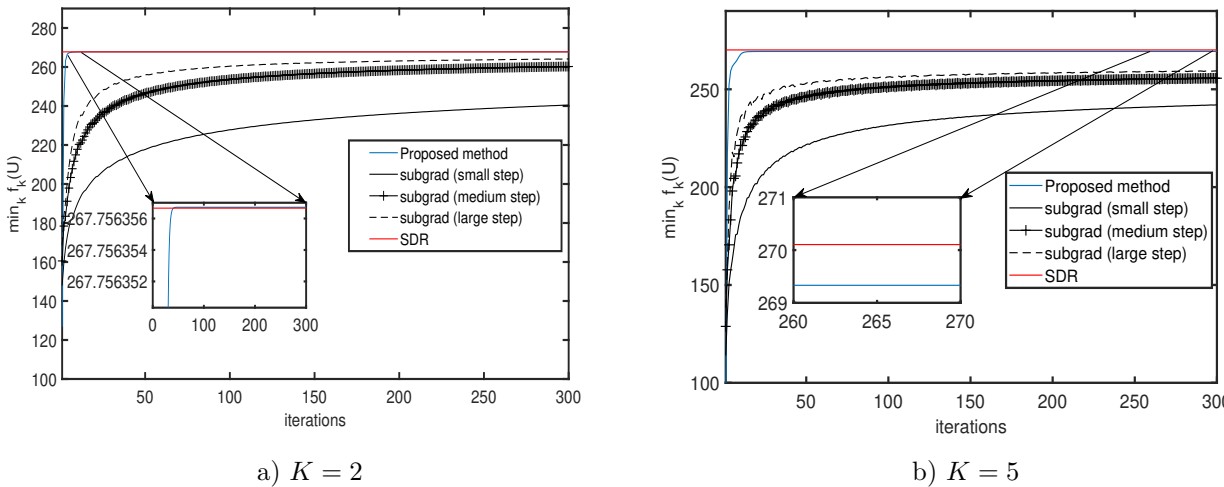

Figure 1: Synthetic data set: Objective vs iteration plots for $n = 10$, $N = 100$, and $r = 4$. In the case of SDR, for the simulation settings in figure 1b, the rank of optimal $\mathbf{P}$ was found to be 6.

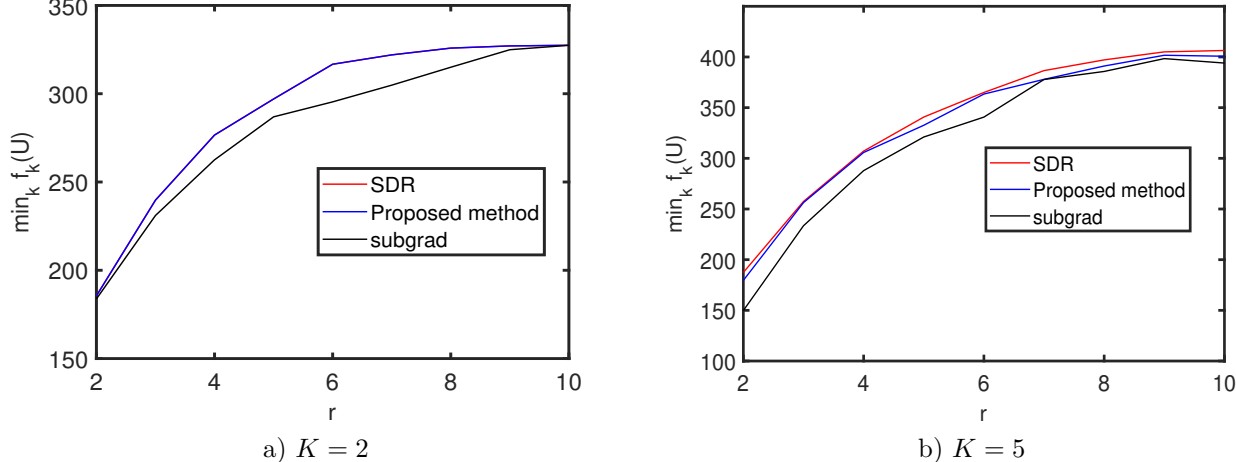

Figure 2: Synthetic data : Objective vs $r$ plots for $n = 10$, $N = 100$.

## 5.1 Synthetic data

The data samples for any class $k$ is generated as $\{\mathbf{y}_i = \mathbf{R}_k^{\frac{1}{2}} \mathbf{e}_i\}$, where $\mathbf{R}_k^{\frac{1}{2}}$ was randomly generated, and the elements of $\mathbf{e}_i$ are i.i.d. random variables sampled from a Gaussian distribution with zero mean and variance one. Let $n = 10, N = 100$, and $r = 4$, and each class has the same number of data samples equal to $\frac{N}{K}$. We first show the objective vs. iteration plots for the three methods in Figure 1 for two different values of $K$. It can be seen from this figure that the proposed FPCA method converges quickly to the SDR upper bound (see the zoom insert in Figure 1a), while the method of Zalcberg & Wiesel (2021) converges slowly to a suboptimal value. For $K > 2$, the SDR upper bound may not be achievable, see for example Figure 1b (zoom insert). For fairness, we ran the method of Zalcberg & Wiesel (2021) for different choices of the step length (small, medium and large) and from Figure 1 one can see that even for different step-lengths the method still converges to suboptimal values. In Figures 2a and 2b, we show the objective values obtained by the three methods for different values of $r$ and $K$. From these figures it can be seen that the proposed FPCA approach always finds a better objective value than the method from Zalcberg & Wiesel (2021) and reaches the SDR bound whenever the latter is achievable.

In Table 1 we show the average computational time (in seconds) of the algorithms.

| $(n, r, K), N = 1000$ | **subgrad** | **Proposed Method** |
|:---:|:---:|:---:|
| (30,2,2) | 1.2 | 2.2 |
| (30,4,2) | 1.8 | 5.8 |
| (30,8,2) | 2.6 | 16.7 |
| (30,16,2) | 4.2 | 48.2 |
| (25,5,5) | 3.1 | 20.2 |
| (50,5,5) | 75.6 | 118.3 |
| (100,5,5) | 450.2 | 740.6 |
| (40,2,2) | 15.1 | 48.6 |
| (40,2,4) | 16.7 | 54.9 |
| (40,2,8) | 20.3 | 60.5 |

Table 1: Average computational time (in seconds) for the algorithms

## 5.2 Credit data

In this simulation we evaluate the performance of the methods on a credit data set from Samadi et al. (2018). First the data is mean centered. The parameters of the credit data are $n = 21$, $K = 2$, $N = 30000$ (with class 1 having 5385 data samples and class 2 having 24615 data samples). In Figure 3a, we plot the objective vs iteration for the proposed FPCA method and the sub-gradient method of Zalcberg & Wiesel (2021). The proposed method quickly converges to the SDR bound. On the other hand, even for a careful choice of the step size, the method of Zalcberg & Wiesel (2021) takes nearly a million iterations to converge to a suboptimal value. In Figure 3b we show the optimal objective values obtained by the three methods for $r$ in the interval $[1, 20]$. It can be seen from this figure that the FPCA method reaches the same objective as SDR.

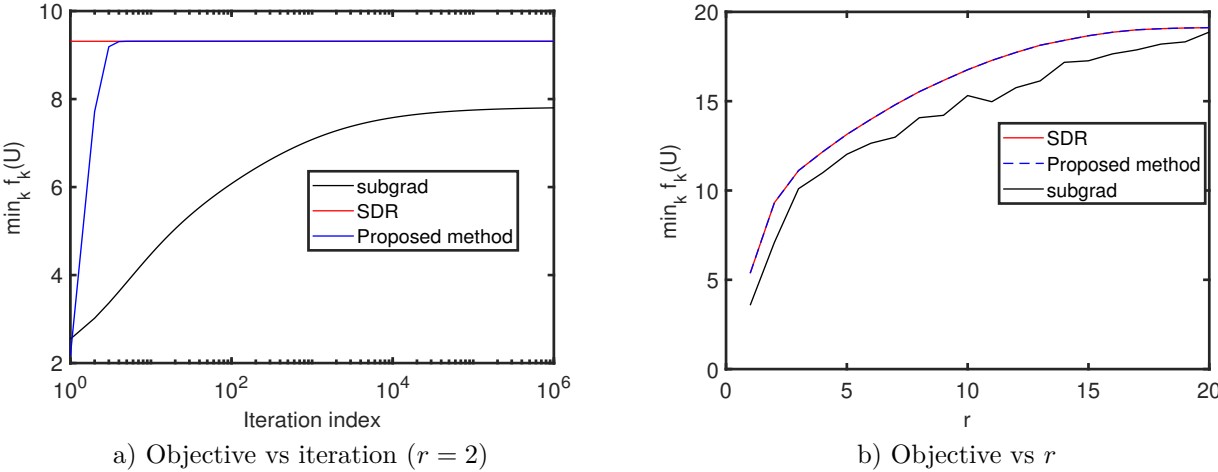

a) Objective vs iteration ($r = 2$)      b) Objective vs $r$

Figure 3: Credit data set with $n = 21$, $N = 30000$, $K = 2$.

## 5.3 LFW data

In this example we compare the performances of the three methods, namely SDR, proposed method and the sub-gradient based method for a high dimensional real data set - labeled faces in the wild (LFW) Huang et al. (2008), which contains the facial images of both men and women ($K = 2$). Each image is of size $42 \times 42$ ($n = 1764$) and there are 2962 images of women and 10270 images of men. In Fig. 4 we show the fair PCA

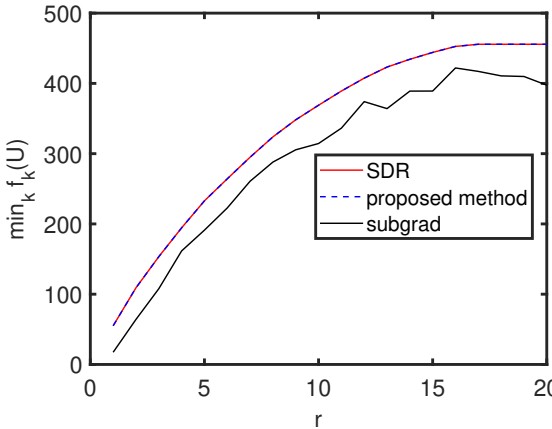

Figure 4: LFW data set with $n = 1764, N = 13232, K = 2$.

objective obtained by the methods for different values of rank ($r$). It can be seen that the proposed method tightly follows the SDR whereas the sub-gradient method is sub-optimal.

### 5.4 Fair robust PCA

In this simulation, we generate the data as in Section 5.1 but now some data samples for randomly chosen classes are corrupted by outliers. The outliers, which are roughly 10% of the total number of samples, are generated from a Gaussian distribution with mean $\alpha$ and variance one. In Figure 5a we plot the objective vs iteration for the proposed FPCA algorithm for ten different initializations. The objective in (5) has several local maxima, which is evident from the plot as the method converges to different local maxima depending on the intialization. In Figure 5b, we vary $\alpha$ in the interval $[0, 20]$ and compute the average normalized error between the estimated subspaces (calculated using 100 Monte-Carlo runs):

$$\text{Normalized subspace error} = \frac{\|\hat{\mathbf{U}}\hat{\mathbf{U}}^T - \tilde{\mathbf{U}}_{\text{FPCA}}\tilde{\mathbf{U}}_{\text{FPCA}}^T\|}{\|\tilde{\mathbf{U}}_{\text{FPCA}}\tilde{\mathbf{U}}_{\text{FPCA}}^T\|} \tag{45}$$

where $\tilde{\mathbf{U}}_{\text{FPCA}}$ denotes the FPCA estimate of the principal components obtained from outlier-free data, and $\hat{\mathbf{U}}$ denotes either $\mathbf{U}_{\text{FPCA}}$ or $\mathbf{U}_{\text{FRPCA}}$ obtained in the presence of outliers. It can be seen from the figure that the FRPCA performs well in the presence of the outliers and has comparatively low errors even in the presence of large outliers.

### 5.5 Fair sparse PCA

In this simulation we generate the data as in Section 5.1 with the choice of parameters: $n = 40, r = 10, N = 200, K = 2$. In Figure 6a, we show the evolution of the objective in (35) vs iterations for $\lambda = 5$. The proposed algorithm monotonically increases the penalized objective as expected. The choice of $\lambda$ determines the sparsness of $\mathbf{U}$. In Figure 6b we show the number of non-zero elements in $\mathbf{U}$ vs $\lambda$ (as expected) when $\lambda$ increases the sparsity of the estimated $\mathbf{U}$ increases.

## 6 Conclusion and Future Work

In this paper we have proposed a new MM algorithm for solving the fair PCA (FPCA) problem. The proposed algorithm is computationally efficient and monotonically increases the FPCA objective at each iteration. Furthermore, the proposed FPCA algorithm does not require the selection of any hyperparameter. We have also proposed two MM algorithms to solve the fair robust PCA and fair sparse PCA problems. Finally, we performed numerical simulations on both synthetic data sets and a real-life data set and compared the performance of the proposed approach with that of two state-of-the-art approaches. In a future work,

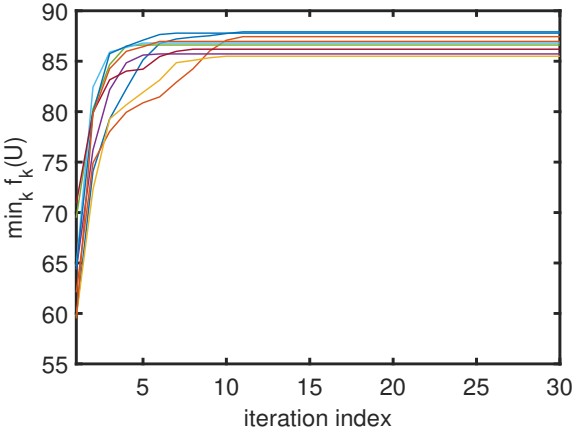
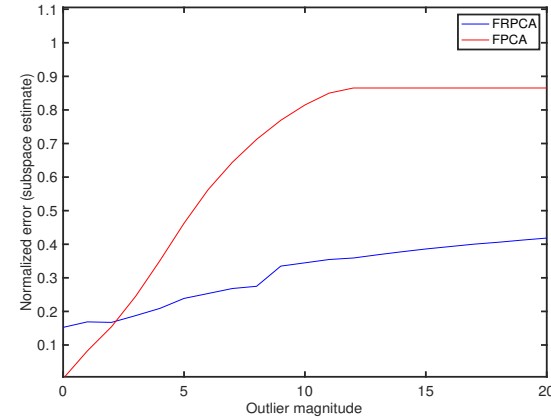

a) Objective vs iteration (for 10 random initializations)    b) Normalized subspace error vs Outlier magnitude ($\alpha$).

Figure 5: Performance of FRPCA algorithm for $n = 10$, $N = 100$, $K = 2$, $r = 4$ and $10\%$ outliers.

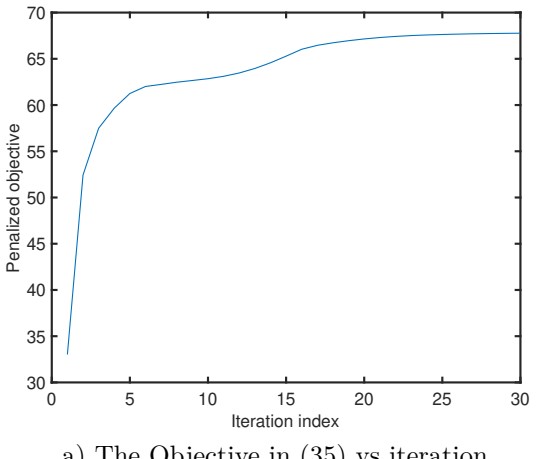
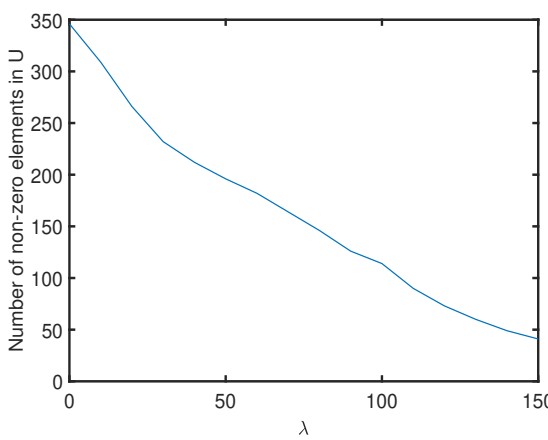

a) The Objective in (35) vs iteration    b) Number of non-zero elements vs $\lambda$

Figure 6: FSPCA: $n = 40$, $r = 10$, $N = 200$, $K = 2$.

we plan to study the convergence properties of the proposed algorithm and derive its convergence rate, with a particular focus on problem (33) and the number of iterations needed to attain practical convergence.

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

## A    Appendix

The minimization problem of FPCA in terms of singular values of $\mathbf{A}$ (given in (29)) is as follows:

$$
\begin{aligned}
\min_{\boldsymbol{\mu}} \quad & 2 \sum_{i=1}^{K} \sigma_i(\mathbf{A}(\boldsymbol{\mu})) + \sum_{k=1}^{K} \mu_k c_k \\
\text{s.t.} \quad & \mu_k \geq 0 \;\;, \sum_{k=1}^{K} \mu_k = 1.
\end{aligned}
\tag{46}
$$

Since the sum of singular values of a matrix is the nuclear norm of the matrix, we have:

$$
\begin{aligned}
\min_{\boldsymbol{\mu}} \quad & 2 \left\| \mathbf{A}(\boldsymbol{\mu}) \right\|_* + \sum_{k=1}^{K} \mu_k c_k \\
\text{s.t.} \quad & \mu_k \geq 0 \;\;, \sum_{k=1}^{K} \mu_k = 1,
\end{aligned}
\tag{47}
$$

where $\left\| \cdot \right\|_*$ denotes the nuclear norm of the matrix. The problem in (47) can be re-written as a semi-definite program (SDP) form Recht et al. (2010) as follows:

$$
\begin{aligned}
\min_{\mathbf{W}_1, \mathbf{W}_2, \boldsymbol{\mu}} \quad & \mathrm{Tr}(\mathbf{W}_1) + \mathrm{Tr}(\mathbf{W}_2) + \sum_{k=1}^{K} \mu_k c_k \\
\text{s.t.} \quad & \mu_k \geq 0 \;\;, \sum_{k=1}^{K} \mu_k = 1, \begin{bmatrix} \mathbf{W}_1 & \mathbf{A}(\boldsymbol{\mu}) \\ \mathbf{A}(\boldsymbol{\mu}) & \mathbf{W}_2 \end{bmatrix} \succcurlyeq 0,
\end{aligned}
\tag{48}
$$

