# OpenReview forum: "Fair principal component analysis (PCA): minorization-maximization algorithms for Fair PCA, Fair Robust PCA and Fair Sparse PCA"
_TMLR — Accepted by TMLR_

### Review · Reviewer_6sKv · 2025-01-01

**Summary Of Contributions:**

In this submission, the authors we propose a new iterative algorithm to solve the fair PCA (FPCA) problem. First, they discuss the general MM approach for a maximization problem, and then the MM approach for max-min problems. The authors present the proposed algorithm for the FPCA problem and discuss an efficient way to implement it. They consider an extension of the proposed method to deal with fair robust PCA and fair sparse PCA problems. The authors also conduct several numerical simulation results on both synthetic and real-life data and compare the proposed method with two state-of-the-art methods.

**Audience:**

Yes

**Claims And Evidence:**

Yes

**Requested Changes:**

There are no significant changes. The authors should add more contents/previous works in the introductory section about the history of the application of MM method to sole the. problems like PCA. Also Forbenius norm is usually denoted as $||A||_F$ instead of $||A||$ for some matrix $A$.

**Strengths And Weaknesses:**

This paper is written with clear structure and high quality language. The authors proposed a new MM algorithm for solving the fair PCA (FPCA) problem. The proposed algorithm is computationally efficient and monotonically increases the FPCA objective at each iteration. The authors also proposed two MM algorithms to solve the fair robust PCA and fair sparse PCA problems. They conducted numerical experiments on both synthetic data sets and a real-life data set and compared the performance of the proposed approach with that of two state-of-the-art approaches. The authors provided details about all the theoretical analysis and mathematical proof and the empirical results from the numerical experiments are consistent with the theoretical results.

One interesting advantage of this submission is that the idea of the application of MM algorithms to solve the PCA problem. This is one novel idea and method. The main idea of the MM algorithm is very similar to the expectation–maximization (EM) algorithm, which is an iterative method to find (local) maximum likelihood or maximum a posteriori (MAP) estimates of parameters in statistical models, Both of them constructed one upper/lower bound of the objective function and then maximize/minimize this upper/lower bound and alternates between these two steps until the convergence error is small enough. This idea is more likely an application of the MM/EM/k-means algorithms into the calculation of the PCA problem. Hence, the main idea of this submission is somehow impressive.

There are no major weakness of this submission.

---

> ### Author Response · Authors · 2025-01-16
> **Response to Review of Paper 3734 by Reviewer 6sKv**
>
> We thank the reviewer for the suggestions and comments on the manuscript. In the Introduction section, we have added some contributions made in solving the PCA problem using MM. The authors of [1] proposed an MM-based algorithm for fast computation of sparse logistic PCA for binary data. An algorithm for the robust sparse PCA using MM combined with orthogonal projection was introduced in [2]. Apart from the aforementioned papers, we have cited several papers on algorithms for PCA/sparse PCA [3-6].
>
> [1] S. Lee and J. Z. Huang, “A coordinate descent mm algorithm for fast computation of sparse logistic
> pca,” Computational Statistics & Data Analysis, vol. 62, pp. 26–38, 2013.
> [2] A. Breloy, S. Kumar, Y. Sun, and D. P. Palomar, “Majorization-minimization on the stiefel manifold
> with application to robust sparse pca,” IEEE Transactions on Signal Processing, vol. 69, pp. 1507–
> 1520, 2021.
> [3] H. Brehier, A. Breloy, M. N. El Korso, and S. Kumar, “Robust and globally sparse Pca via
> majorization-minimization and variable splitting,” in ICASSP 2023-2023 IEEE International Con-
> ference on Acoustics, Speech and Signal Processing (ICASSP). IEEE, 2023, pp. 1–5.
> [4] Q. Wang, Q. Gao, X. Gao, and F. Nie, “ℓ2,p-norm based pca for image recognition,” IEEE Transac-
> tions on Image Processing, vol. 27, no. 3, pp. 1336–1346, 2017.
> [5] R. Wang, F. Nie, X. Yang, F. Gao, and M. Yao, “Robust 2DPCA With Non-greedy ℓ1-Norm Max-
> imization for Image Analysis,” IEEE transactions on cybernetics, vol. 45, no. 5, pp. 1108–1112,
> 2014.
> [6] C. Zhang, F. Nie, and S. Xiang, “A general kernelization framework for learning algorithms based
> on kernel PCA,” Neurocomputing, vol. 73, no. 4-6, pp. 959–967, 2010.

---

### Review · Reviewer_KaBw · 2025-01-03

**Summary Of Contributions:**

This paper proposes a new iterative algorithm based on the minorization-maximization (MM) approach to solve the fair principal component analysis (FPCA) problem. The proposed algorithm is computationally efficient, monotonically increases the objective at each iteration, and does not require any hyperparameter tuning. The authors also present extensions of the algorithm to handle robust PCA (FRPCA) and sparse PCA (FSPCA) formulations of the fair PCA problem. Numerical experiments on synthetic and real-world datasets show the effectiveness of the proposed approach compared to existing state-of-the-art methods.

**Audience:**

Yes

**Broader Impact Concerns:**

There aren't any major ethical implications of the work.

**Claims And Evidence:**

Yes

**Requested Changes:**

I suggest the authors to address the following comments:

1. Please use a unified equation referencing style: equation 4 vs (4), equation 6 vs (6).
2. The definition of $\ell$-1 norm of a matrix below (6) should be placed at just after (5).
3. The definitions in (9) and (10) can be placed before (8) for consistency.
4. Please consider summarizing the ``interesting properties'' for (23) into a lemma and put the following proof into a former proof.
5. (28) can be formulated as an SDP, what is the SDP? This is necessary since we need to see the SDP to understand the discussion of computational complexity.
6. Explicitly state the dependency $A(\mu)$, otherwise it is quite misleading to ignore $\mu$.
7. The problem (33) has to be solved for multiple times, and usually only a few iterations are needed. Is there any theoretical result including the iteration complexity to support this point?
8. Please also include some discussions regarding the convergence properties (in more details) and the computational cost with memory consumption.
9. It is also suggested to include comparison of computational time in the experiments?
10. Are all data set used mean centered?
11. What is the Section V.A? Please correct similar typos.
12. Please provide more details on how to get the reformulation (24)?

**Strengths And Weaknesses:**

## Strengths
1. The paper is well-written, easy to follow, the ideas are explained clearly, and the algorithms are easy to understand.
2. The paper contains extensive numerical results to support the findings.

## Weakness
1. While the writing style is clear, the technical content is not well-organized.
2. The paper mentions several times that the MM algorithm is convergence, but it didn't formally state these convergence results with proper citations.

Overall, the work presents a valuable contribution to the field of FPCA and related extensions, which is of interest to the TMLR audience.

---

> ### Author Response · Authors · 2025-01-16
> **Response to Review of Paper 3734 by Reviewer KaBw**
>
> We thank the reviewer for the comments. We have addressed the comments in the following manner:
>     1. We have revised the manuscript and made sure that the referencing style is consistent throughout the paper. Thanks for this observation.
>     2. We have now introduced the $\ell_1$ norm of the matrix after (5).
>     3. Thank you for the suggestion. The requested sequence is now followed in the updated manuscript.
>     4. We have given a new lemma for (23) in the updated manuscript.
>     5. The problem in (28) in the manuscript is as follows:
> \begin{array}{ll}
> \min\limits_{\boldsymbol{\mu}} {2\sum\limits_{i=1}^{K}\sigma_{i}(\mathbf{A}(\boldsymbol{\mu}))+\sum\limits_{k=1}^{K}\mu_{k}c_{k}}\\
> \textrm{s.t.}~\mu_{k} \geq 0, \sum\limits_{k=1}^{K}\mu_{k}=1.
> \end{array}
>
> Since the sum of singular values of a matrix is the nuclear norm of the matrix, we have:
> \begin{array}{ll}
> \min\limits_{\boldsymbol\mu} &{2\lVert\mathbf{A}(\boldsymbol{\mu})\rVert_{*}+\sum\limits_{k=1}^{K}\mu_{k}c_{k}}\\
> \textrm{s.t.}&{\mu_{k} \geq 0, \sum\limits_{k=1}^{K}\mu_{k}=1},
> \end{array}
>
> where $\lVert \cdot \rVert_{*}$ denotes the nuclear norm of the matrix. The above problem can be written in SDP form [1] as the following:
>
> \begin{array}{ll}
> \min\limits_{\mathbf{W}_1, \mathbf{W}_2, \boldsymbol{\mu}} &\textrm{Tr}(\mathbf{W}_1) + \textrm{Tr}(\mathbf{W}_2) + \sum\_{k=1}^{K} \mu_k c_k
> \end{array}
>
> \begin{array}{ll}
> \textrm{s.t.} & {\mu_{k} \geq 0, \sum\limits_{k=1}^{K}\mu_{k}=1}, \begin{bmatrix} \mathbf{W}_1 & \mathbf{A}(\boldsymbol{\mu}) \\\
> \mathbf{A}(\boldsymbol{\mu}) &  \mathbf{W}_2 \end{bmatrix} \succcurlyeq {0},
> \end{array}
>
>  We included a reference to [1] in the revised paper.
>    6. The dependency of $\mathbf{A}(\boldsymbol{\mu})$ is stated in the updated manuscript to avoid confusion.
>    7. There is hardly any theoretical result on this aspect. Nonetheless, from our numerical simulation experience, we have observed that solving (33) typically requires around $10$ iterations (even for problem dimension $n = 100$) to achieve convergence (as can also be seen from the figures in section $5$).
>    8.  To claim convergence to a stationary point would require additional assumptions and further analysis. What we can say for sure is that the sequence of the objective function evaluated at the iterates will be monotonically increasing at each iteration, and since the objective is bounded above, thus we can only claim convergence in terms of the objective function.
>   Regarding the memory requirement of the proposed algorithm, the matrices $\{\mathbf{A}_{k}\} ({k=1\cdots K})$ which need to be stored require $\mathcal{O}(Krn)$ memory words, and the memory requirement of the SDP is around $\mathcal{O}\big((n+r)^{2}\big)$, thus the total memory requirement is on the order of $\mathcal{O}\big((n+r)^{2}\big)$.
>   9. For comparing the computational time of the algorithms, we perform the simulation and tabulate the computation time in the following table:
>
> | $(n,r,K), N=1000$ | subgrad  | Proposed Method |
> |-----------|-----------|---------|
> | (10,2,2) | 0.3 | 1.3 |
> | (50,5,3) | 60.8 | 112.2 |
> | (100,8,5) | 500.1 | 800.6 |
>
> **Average computational time (in seconds) for the algorithms.**
>
>   10. Yes, the data samples are mean-centered.
>   11. The references to the section are now corrected and consistent in the updated manuscript.
>   12.
>
> \begin{align}
>     \min_k ~ g_k(\mathbf{U}) = &\min_{\boldsymbol{\mu}}~ \sum_{k=1}^K \mu_k g_k(\mathbf{U})\\
>     &\textrm{s.t.} ~\mu_k \ge 0, \boldsymbol{\mu}^{T}\mathbf{1}=\mathbf{1}.
> \end{align}
>
>  Solving the problem on R.H.S. of the above equation, the elements of the vector $\boldsymbol{\mu}$ are obtained such that $\mu_k$ corresponding to the minimum value of $g_k(\mathbf{U})$ is $1$ and rest $0$, giving the minimum value of $g_k(\mathbf{U})$.
>
> [1] B. Recht, M. Fazel, and P. A. Parrilo, “Guaranteed minimum-rank solutions of linear matrix equations
> via nuclear norm minimization,” SIAM review, vol. 52, no. 3, pp. 471–501, 2010.

---

### Review · Reviewer_YYj6 · 2025-01-09

**Summary Of Contributions:**

The paper developed an MM method to solve the fair PCA problem proposed in [Samadi et al. 2018]. The authors further proposed two variants of the fair PCA problem, namely fair robust PCA (FRPCA) and fair sparse PCA (FSPCA) by replacing the (loosely speaking) quadratic objective function with the L1 counterpart and adding the (negative) L1 norm of the optimization variable to the objective function. They show that the mentioned MM method can be straightforwardly extended to tackle those newly proposed problems. They perform several numerical experiments to solve those three problems, and the results are quite sound.

**Audience:**

Yes

**Claims And Evidence:**

Yes

**Requested Changes:**

Those recommendations are mainly to improve the paper:

1) Please mention that the crucial requirement of MM is that the surrogates are easy to maximise.

2) Some claims about the MM should be made more accurate. On page 4, it says "convergence to a stationary point." But as far as I know, in general, MM does not converge to any stationary points.  The limiting set should be "no-progress" points [Lange21] where MM can not improve the solution anymore, it does not mean the gradient of the objective function vanishes there. Furthermore, there are cases where MM -- at its limit -- jumps around different no-progress points, so it is not convergent [see Lange21]. The objective function of fair PCA has some structure, I would expect a stronger conclusion to hold. The authors cited [Zhao & Palomar 2016], please indicate precisely which theorems in [Zhao & Palomar 2016] apply here.

[Lange21] Lange, Kenneth, et al. "Nonconvex optimization via MM algorithms: Convergence theory." arXiv preprint arXiv:2106.02805 (2021).

3) All the identity matrices should be associated with their dimensions.

4) The first equation of the Proof in 5 seems redundant, I suggest omitting it to make the proof neater.

5) Right after equation (26), please recall that $c_k$ does not depend on $U$ so we can safely ignore the second term.

6) In Algorithm 1 (and other algorithms as well), please explain why the stopping condition is well-defined. For example, as I mentioned, there are cases where MM jumps between no-progress points at its limit, therefore $U^{l+1}$ is not close to $U^t$.

7) In the complexity discussion, a citation is needed for the complexity of $(n+r)^{4.5}$ of the SDP solver.

8) Please give more explanation for the choice of the surrogate in (41). Is it a common choice for L1?

9) The last claim in subsection 5.1. is not obvious to me: "The proposed FPCA approach always finds a better objective value than the method from Zalcberg & Wiesel (2021) and reaches the SDR bound whenever the latter is achievable." I believe this claim is not sufficiently supported.

10) In section 5.4., the description of the added noise is not clear. Please provide more explanations.

11) The claim: "Unlike the FPCA and the FSPCA case, the objective in (5) has several local maxima" implies the objective of FPCA and FSPCA do not have any local maxima. This does not seem to be the case because the objectives of FPCA and FSPCA are nonconvex. Please provide explanations.

12) Where is the SDR curve in Figure 2(a)?

**Strengths And Weaknesses:**

**Strengths:**

- The paper is very well-written with sufficient background and discussion.
- The proposed MM method is robust and natural for the max-min type of the objective function of fair PCA.
- As a strength of MM, it is parameter-free, which means no tuning is needed.
- The math seems clear and sound to me, I did check some proofs and arguments and they seem correct.
- Numerical experiments are sound: they do show that the MM method (which is exact in terms of rank) closely matches the method proposed in  [Samadi et al. 2018] (which is -- in general -- not exact in terms of rank due to the parametrization and relaxation) and outperforms the subgradient method.

**Weaknesses:**

- The paper does not contribute any added insight in terms of convergence of the proposed method, but rather, it uses general convergence of the MM.
- Iteration complexity is of order $n^{4.5}$ where $n$ is the data dimension. Although they managed to propose an iterative cheaper version with a complexity of $n^3$, this version is sub-optimal (hard to say theoretically to stop the inner-loop algorithm).
- Plotting objective against iterations may be unfair for the subgradient method since the iteration complexity of the MM is higher.

---

> ### Author Response · Authors · 2025-01-16
> **Response to Review of Paper 3734 by Reviewer YYj6**
>
> We thank the reviewer for the recommendations/suggestions. Below is our point-by-point response.
>    1. Following your suggestion, we state in Section 2.1 that the surrogate function ideally should follow the shape of the objective function and be easy to maximize.
>    2. To claim convergence to a stationary point would require additional assumptions and further analysis. We have now removed that claim from the manuscript. We can say for sure that the sequence of the objective function evaluated at the iterates will be monotonically increasing at each iteration, and since the objective is bounded above, thus we can only claim convergence in terms of the objective function. We have now restated the aforementioned point in the manuscript and have also cited [1].
>  3. The updated manuscript has the notations of identity matrices with their respective dimensions.
>  4. Thank you for your comment. We have removed the redundant equation.
>  5. The second term in (25) cannot be left out, as $\boldsymbol{\mu}$ is still a minimization variable.
>  6. We agree with the reviewer that the stopping criterion based on relative change in $\mathbf{U}$ is not a good choice. We have now replaced it with the stopping criterion in terms of the objective function $\big(\frac{f(\mathbf{U}^{t+1}) - f(\mathbf{U}^t)}{f(\mathbf{U}^t)}\big)$.
> 7. Yes, the available SDP solvers have the computational complexity of $\mathcal{O}(n^{4.5})$ [2-3]. We have also cited the references in the updated manuscript.
> 8. Yes, it is a common choice of surrogate for the $\ell_1$ norm.
> 9. The claim in subsection 5.1 is based on numerical simulations performed for the evaluation of the proposed FPCA algorithm on synthetically generated data. It is supported using the objective vs iteration plots in Figure 1 and objective vs $r$ plots in Figure 2 of the manuscript.
>     * It can be seen that the objective attained by the FPCA method on convergence is higher than that by the method from Zalcberg \& Wiesel (2021).
>     * Also, for $K=2$, the objective attained by the FPCA method converges to that of SDR method (as seen from the zoom in Figure 1a and Figure 2a).
>     * For $K=5$, as discussed in section 5, due to the relaxation involved, the objective attained by SDR is the upperbound and is not achievable, and so, the objective attained by the FPCA method is lower than that of SDR (as shown in the zoom plot in Figure 1b and Figure 2b).
> 10. In section 5.4, the data for FRPCA is generated as described in section 5.1. The outliers are drawn randomly from a Gaussian distribution of mean $\alpha$ and variance $1$. The number of outliers are $10\%$ of the data samples.
> 11. The line "Unlike the FPCA and the FSPCA case, the objective in (5) has
>     several local maxima" was misleading and therefore we have removed it from the manuscript.
> 12.  As discussed, the proposed FPCA method achieves the SDR bound for $K<=2$, and so in Figure 2a (for $K=2$), the plots of FPCA and SDR objectives are identical.
>
> [1] K. Lange, J.-H. Won, A. Landeros, and H. Zhou, “Nonconvex optimization via mm algorithms:
> Convergence theory,” arXiv preprint arXiv:2106.02805, 2021.
> [2] C. Shen, A. Welsh, and L. Wang, “PSDBoost: Matrix-generation linear programming for positive
> semidefinite matrices learning,” Advances in Neural Information Processing Systems, vol. 21, 2008.
> [3] Y. Zheng, Y. Yan, S. Liu, X. Huang, and W. Xu, “An Efficient Approach to Solve the Large-Scale
> Semidefinite Programming Problems,” Mathematical Problems in Engineering, vol. 2012, no. 1, p.764760, 2012.

---

### Decision · Action_Editor_ki62 · 2025-02-19

**Recommendation:** Accept with minor revision

**Comment:**

While the authors have addressed all reviewers' concerns, some of the requested changes from reviewer KaBw should be better implemented in the paper, to improve clarity and claims/evidence:
- question 4: Lemma 2 proof is missing the Latex commands begin{proof} and end{proof}
- question 5: Please include the SDP in the appendix.
- question 7: Please mention that while problem (33) is solved in less than 10 iterations experimentally for problems of size up to n=100, there is no theoretical result on this respect.
- question 9: Please include a table of the computational time. Varying only one parameter (n, r or K) while keeping the rest fixed might be a way to do a more proper comparison of the computational time.
- question 12: Please add this small explanation to the paper for didactical purposes.

Finally, it is very important to add some ideas for future work in Section 6. Perhaps question 7 from reviewer KaBw is one candidate, but other ideas will be highly welcome by TMLR's readership.

**Audience:**

All reviewers agree that the manuscript has an audience. Fair PCA is definitely relevant to at least a some individuals in the machine learning community and the TMLR readership. The audience might be relatively smaller when compared to main-trend machine learning problems, but it is important to let the community know of these developments, and to increase the diversity of topics studied by the community.

**Claims And Evidence:**

All reviewers agree that the manuscript presents appropriate evidence for its claims. The manuscript proposes single and efficient iterative algorithms for fair PCA, based on minorization-maximization. Extensions for the robust and sparse case are also provided. Some properties are proven formally through lemmas. Experimental evidence is also provided on synthetic and real world data.

---

> ### Author Response · Authors · 2025-02-24
> **Response to decision by Action Editor ki62**
>
> We have addressed the concerns and made the requested changes in the revised manuscript (uploaded as camera ready version).